# Local Latent Space Bayesian Optimization over Structured Inputs

**Natalie T. Maus**
Department of Computer and Information Science
University of Pennsylvania
Philadelphia, PA
nmaus@seas.upenn.edu

**Haydn T. Jones**
Los Alamos National Laboratory
Los Alamos, NM
hjones@lanl.gov

**Juston S. Moore**
Los Alamos National Laboratory
Los Alamos, NM
juston@lanl.gov

**Matt J. Kusner**
Centre for Artificial Intelligence,
University College London
London, UK
m.kusner@ucl.ac.uk

**John Bradshaw**
Department of Chemical Engineering,
Massachusetts Institute of Technology
Cambridge, MA
jbrad@mit.edu

**Jacob R. Gardner**
Department of Computer and Information Science
University of Pennsylvania
Philadelphia, PA
jacobrg@seas.upenn.edu

## Abstract

Bayesian optimization over the latent spaces of deep autoencoder models (DAEs) has recently emerged as a promising new approach for optimizing challenging black-box functions over structured, discrete, hard-to-enumerate search spaces (e.g., molecules). Here the DAE dramatically simplifies the search space by mapping inputs into a continuous latent space where familiar Bayesian optimization tools can be more readily applied. Despite this simplification, the latent space typically remains high-dimensional. Thus, even with a well-suited latent space, these approaches do not necessarily provide a complete solution, but may rather shift the structured optimization problem to a high-dimensional one. In this paper, we propose LOL-BO, which adapts the notion of trust regions explored in recent work on high-dimensional Bayesian optimization to the structured setting. By reformulating the encoder to function as both an encoder for the DAE globally and as a deep kernel for the surrogate model within a trust region, we better align the notion of local optimization in the latent space with local optimization in the input space. LOL-BO achieves as much as 20 times improvement over state-of-the-art latent space Bayesian optimization methods across six real-world benchmarks, demonstrating that improvement in optimization strategies is as important as developing better DAE models.

## 1  Introduction

Many challenges across the physical sciences and engineering require the optimization of *structured objects*. For example, in the drug design setting, it is common to seek a molecule that maximizes some property, e.g., binding affinity for a target protein [17], or similarity to a known drug molecule

36th Conference on Neural Information Processing Systems (NeurIPS 2022).

under some metric [3]. These structured optimization problems are particularly challenging, as the objective function is now typically over a large combinatorial space of discrete objects.

These challenging optimization problems have motivated a large amount of very recent interest in methods that perform *Bayesian optimization (BO) over latent spaces*. Using drug design as a running example, a deep auto-encoder (DAE) is first trained on a large set of unsupervised molecules. Bayesian optimization is then run as normal in the continuous latent space of the autoencoder to produce candidate latent codes. These candidate latent codes are then decoded back to molecules, and the objective function is evaluated. Bayesian optimization then proceeds as normal, with the surrogate model trained on a dataset of latent codes and their resulting objective values after decoding.

Rapid progress has been made in this emerging topic. In large part, this has been to improve the DAE model. The early work of Kusner et al. [28] and Jin et al. [21] investigated domain specific autoencoder architectures that much more reliably produced valid molecules. The latest work of Tripp et al. [43] and Grosnit et al. [14] investigate methods to inject supervised signal from objective function evaluations into the latent space, enabling better surrogate modeling. However, the *optimization* component of latent space Bayesian optimization remains under-explored: indeed, it is common practice to directly apply standard Bayesian optimization with expected improvement once a latent space has been constructed (all of the above cited methods do this). We conjecture that, despite the ability of a deep encoder to dramatically simplify a structured input space, optimization remains challenging due to the high-dimensionality of the extracted latent space.

In this paper, we propose to jointly co-design the latent space model and a local Bayesian optimization strategy. Concretely, we make the following contributions:

1. We develop local latent Bayesian optimization, LOL-BO, for the structured optimization setting, addressing the inherent mismatch between the notion of a trust region in the latent space and a trust region in the original structured input space.

2. We derive a semisupervised DAE model that better satisfies the implicit assumptions of trust region methods by performing joint variational inference over the DAE and sparse GP surrogate model (this model is similar in spirit to Kingma et al. [25], Eissman et al. [6]).

3. We propose a deep autoencoder architecture using transformer layers over the recently proposed SELFIES representation for molecules [27]. We find that the SELFIES string representation provides significant improvements to optimization performance.

4. We show the potential of co-adapting latent space modeling with local Bayesian optimization in an empirical evaluation of LOL-BO against top-performing baselines, achieving up to $22\times$ improvements in objective value on commonly used benchmark tasks over prior latent space BO approaches.

## 2 Background

In black-box optimization, we seek to find the minimizer of a function, $\arg\min_{\mathbf{x}\in\mathcal{X}} f(\mathbf{x})$. Commonly, $f(\mathbf{x})$ is assumed to be expensive to evaluate and unknown (i.e., a "black box"). It is therefore desirable to use fewer function evaluations to achieve a given objective value.

Most classically, this problem has been considered in the setting where the search space $\mathcal{X}$ is continuous (e.g., $\mathcal{X} \subseteq \mathbb{R}$). However, many applications across the natural sciences and engineering require optimizing over discrete and structured input spaces. *De novo* molecule design using machine learning has received particular attention, where $\mathcal{X}$ is a search space of small organic molecules and $f(\mathbf{x})$ is some desirable property to optimize [31].

**Bayesian optimization** [37, 30, 41] is a framework for solving these black-box optimization problems, most commonly when $\mathcal{X}$ is a continuous, real-valued search space. A surrogate model is trained on a set of inputs and corresponding objective values $\mathcal{D}_\ell = [\mathbf{x}_i, y_i]_{i=1}^n$. This surrogate model is used to suggest candidate inputs $\hat{\mathbf{x}}$ to query next. The candidates are then labeled, added to the dataset, and the process repeats. Gaussian process models $f(\mathbf{x}) \sim \mathcal{GP}(m(\mathbf{x}), k(\mathbf{x}, \mathbf{x}'))$ are commonly used as surrogate models, as access to calibrated predictive (co-)variances enables search strategies that trade off exploration and exploitation.

**Deep autoencoder models.** A deep autoencoder model consists of an encoder $\Phi : \mathcal{X} \to \mathcal{Z}$ that maps from a structured and discrete space (e.g., the space of molecules) $\mathcal{X}$ to a continuous, real-valued *latent space* $\mathcal{Z}$, and a decoder that does the reverse $\Gamma(\cdot) : \mathcal{Z} \to \mathcal{X}$. A variational autoencoder [24] uses a probabilistic form of these two functions, a variational posterior $\Phi(\mathbf{Z} \mid \mathbf{X})$ and a likelihood $\Gamma(\mathbf{X} \mid \mathbf{Z})$. The parameters $\theta_\Phi, \theta_\Gamma$ of $\Phi$ and $\Gamma$ are trained by maximizing the evidence lower bound (ELBO):

$$\mathcal{L}_{\text{VAE}} = \text{ELBO}(\theta_\Phi, \theta_\Gamma) = \mathbb{E}_{\Phi(\mathbf{Z}|\mathbf{X})}[\log \Gamma(\mathbf{X} \mid \mathbf{Z})] - \text{KL}(\Phi(\mathbf{Z} \mid \mathbf{X}) \,||\, p(\mathbf{Z}))$$

Commonly, the (amortized) variational posterior $\Phi(\mathbf{Z} \mid \mathbf{X})$ and the prior $p(\mathbf{Z})$ are chosen to be Gaussian, which leads to a convenient analytic expression for the KL divergence. Note that, with respect to specific optimization tasks we might want to perform over the structured space $\mathcal{X}$, the ELBO as defined above is unsupervised.

**Latent space Bayesian optimization.** Latent space Bayesian optimization seeks to leverage the representation learning capabilities of deep autoencoders models (DAEs) to reduce a discrete, structured search space $\mathcal{X}$ to a continuous one $\mathcal{Z}$ [43, 5, 12, 14, 6, 22]. Because the ELBO above is unsupervised with respect to an optimization task, a large set of unlabeled inputs $\mathcal{D}_u \subseteq \mathcal{X}$ is first used to train the DAE. A Gaussian process prior is again placed over the objective function, but here as a function of the *latent codes*, $f(\mathbf{z}) \sim \mathcal{GP}(m(\mathbf{z}), k(\mathbf{z}, \mathbf{z}'))$. To train the GP initially, we collect a small set of $n$ inputs $\mathcal{D}_\ell \subseteq \mathcal{X}, n \ll |\mathcal{D}_u|$ with known labels $\mathbf{y}_\ell$. The inputs are passed through the encoder to obtain latent representations $\mathbf{Z} = [\mathbf{z}_i]_{i=1}^n$, and the surrogate model is fit to the dataset $[(\mathbf{z}_i, y_i)]_{i=1}^n$. Standard BO strategies may then be applied using this surrogate model to select a candidate latent vector $\hat{\mathbf{z}}$, which is then decoded back to an input $\hat{\mathbf{x}}$ and evaluated.

## 3 Methods

Recent latent space Bayesian optimization (LS-BO) approaches have focused on improving the architecture of the VAE used to compress the search space [28, 21], and on reorganizing the subsequent latent space so that it is more amenable to BO [43, 14, 6]. However, it is commonplace to apply standard BO with expected improvement once the latent space has been constructed, despite the fact that this latent space often remains high-dimensional [21], on the order of tens or hundreds of dimensions. In this paper, we explore whether lessons learned in the high-dimensional BO setting can be adapted to improve optimization in high-dimensional latent spaces.

### 3.1 Local Bayesian optimization – Fixed latent space

One approach to adapting high-dimensional BO strategies is to apply them in the extracted latent space $\mathcal{Z}$. Because $\mathcal{Z}$ is a synthetic latent space, it's unlikely that the objective function decomposes additively over this space without explicitly encouraging this [23, 9, 47]. Likewise, $\mathcal{Z}$ is already a compression of the much larger $\mathcal{X}$, so methods that exploit low-dimensional substructure may not be effective [48, 32, 7, 11].

*Local Bayesian optimization* is a strategy that avoids over-exploration in these search spaces *without* simplifying assumptions about the objective. Commonly this is done by restricting the search in each iteration to a small region. For example, Eriksson et al. [8] proposes TuRBO-M, an optimization algorithm which maintains $M$ local optimization runs, each of which is limited to search within its own hyper-rectangular *trust region*. Each trust region is centered at the current best input found by a given run. Since Eriksson et al. [8] found that TuRBO-1 is competitive or even better than TuRBO-m where $m > 1$ on most tasks, in this paper we focus only on TuRBO-1. Following Eriksson et al. [8], we will henceforth use TuRBO as short-hand for TuRBO-1.

Adapting trust regions to LS-BO is straightforward. Suppose we have a VAE with encoder $\Phi(\cdot)$ and decoder $\Gamma(\cdot)$ trained on an unsupervised dataset $\mathcal{D}_u$ of structured inputs, and a supervised set $\mathbf{Z}$ of latent codes corresponding to labeled inputs $\mathcal{D}_\ell$ and their objective values $\mathbf{y}$. Denote by $\hat{\mathbf{z}}$ the *incumbent* latent code corresponding to the input $\hat{\mathbf{x}}$ with best observed objective value $\hat{y}$ so far.

The trust region is an axis aligned hyper-rectangular subset of the latent space $\mathcal{T}_{\hat{z}} \subseteq \mathcal{Z}$, centered at the incumbent latent vector $\hat{\mathbf{z}}$. It has a base side length $L$, with the actual side length for each dimension rescaled according to its lengthscale $\lambda_i$ in the GP's ARD covariance function, $L_i = \frac{\lambda_i L}{(\prod_{j=1}^d \lambda_j)^{1/d}}$.

Candidate latent vectors to evaluate next are restricted to be selected from $\mathcal{T}_{\hat{z}}$. Following Nelder and Mead [33], Eriksson et al. [8], the base trust region length is halved after $\tau_{\text{fail}}$ iterations where no progress is made and doubled after $\tau_{\text{succ}}$ iterations in a row where progress is made. When a new incumbent $\hat{\mathbf{z}}'$ is found with $\hat{y}' < \hat{y}$, the trust region is recentered at $\hat{\mathbf{z}}'$.

## 3.2 Local Bayesian optimization – Adaptive latent space

By defining "locality" in terms of a small rectangular region centered at the incumbent, trust region methods implicitly assume smoothness, that points close to the center have highly correlated objective values with it. In the standard BO setting with GP surrogates, this assumption matches that made by the GP, as many covariance functions measure covariance as a function of distance.

However, the latent space $\mathcal{Z}$ of a DAE is only trained so that a latent vector $\mathbf{z} = \Phi(\mathbf{x})$ reconstructs the original input when passed through the decoder, $\Gamma(\mathbf{z}) \approx \mathbf{x}$. Therefore, it is possible that small changes to the current incumbent *latent vector* $\hat{\mathbf{z}}$ may result in large changes to the corresponding *input* $\hat{\mathbf{x}}$, violating the assumptions of both the GP surrogate model and the trust region method.

**Adapting the latent space to the GP prior.** We propose to solve this mismatch by performing inference jointly over both the Gaussian process surrogate defined over the latent space $p(\mathbf{f} \mid \mathbf{Z})$ and the Bayesian VAE model $p(\mathbf{X}, \mathbf{Z}) = p(\mathbf{X} \mid \mathbf{Z})p(\mathbf{Z})$, while assuming in the prior conditional independence between $\mathbf{f}$ and $\mathbf{X}$ given $\mathbf{Z}$:

$$p(\mathbf{f}, \mathbf{X}, \mathbf{Z}) = p(\mathbf{f} \mid \mathbf{Z})\Gamma(\mathbf{X} \mid \mathbf{Z})p(\mathbf{Z})$$

By performing inference over $\mathbf{f}$ and $\mathbf{Z}$ jointly in the above model end-to-end, we encourage the latent space to organize in a way that matches the assumptions of the GP prior $p(\mathbf{f} \mid \mathbf{Z})$.

**Relation to prior work.** Learning the VAE and supervised model jointly in a semisupervised fashion is related to prior work on semisupervised VAE modelling as first proposed in Kingma et al. [25]. This was adapted for BO in Eissman et al. [6], who use an auxiliary supervised neural network to inject supervision before fitting the GP. Here we consider joint variational inference over the VAE and a sparse GP, directly encouraging the latent space to satisfy the assumptions of the GP prior. In the global BO setting, Siivola et al. [40] finds semisupervised adaptation of VAEs to not be critical; however, in the local setting, we experimentally find this results in significant improvement due to the assumptions underlying trust region methods.

**Sparse GP augmentation.** Following Hensman et al. [15], the above prior $p(\mathbf{f} \mid \mathbf{Z})$ can be augmented with a global set of learned *inducing variables* $\mathbf{u}$ representing latent function values at a set of *inducing locations* $\mathbf{V}$, $p(\mathbf{f} \mid \mathbf{Z}) \rightarrow p(\mathbf{f} \mid \mathbf{u}, \mathbf{Z}, \mathbf{V})p(\mathbf{u} \mid \mathbf{V})$. This results in the following full joint density over the latent function $\mathbf{f}$, structured inputs $\mathbf{X}$, latent representation $\mathbf{Z}$, objective values $\mathbf{y}$, and inducing variables $\mathbf{u}$ (ommitting dependencies on inducing locations $\mathbf{V}$ for compactness):

$$p(\mathbf{y}, \mathbf{X}, \mathbf{f}, \mathbf{u}, \mathbf{Z}) = p(\mathbf{y} \mid \mathbf{f})p(\mathbf{f} \mid \mathbf{u}, \mathbf{Z})p(\mathbf{u})\Gamma(\mathbf{X} \mid \mathbf{Z})p(\mathbf{Z}) \tag{1}$$

**ELBO derivation.** Our goal is to derive an evidence lower bound (ELBO) for the above model to perform variational inference over both the inducing variables $\mathbf{u}$ and VAE latent variables $\mathbf{Z}$. The log evidence $p(\mathbf{y}, \mathbf{X})$ for this model is $\log p(\mathbf{y}, \mathbf{X}) = \log \int p(\mathbf{y}, \mathbf{X}, \mathbf{f}, \mathbf{u}, \mathbf{Z})d\mathbf{f}d\mathbf{u}d\mathbf{Z}$. To aid in lower bounding the log evidence, we introduce a variational distribution $q(\mathbf{u}, \mathbf{Z}) = q(\mathbf{u})\Phi(\mathbf{Z} \mid \mathbf{X})$. This is a factored joint variational distribution over the inducing values $\mathbf{u}$ as in [15, 42] and over the latent space of the encoder $\mathbf{Z}$.

$$\log p(\mathbf{y}, \mathbf{X}) = \log \int p(\mathbf{y}, \mathbf{X}, \mathbf{f}, \mathbf{u}, \mathbf{Z})d\mathbf{f}\frac{q(\mathbf{u})}{q(\mathbf{u})}d\mathbf{u}\frac{\Phi(\mathbf{Z} \mid \mathbf{X})}{\Phi(\mathbf{Z} \mid \mathbf{X})}d\mathbf{Z}$$

$$= \log \mathbb{E}_{\Phi(\mathbf{Z}|\mathbf{X})}\left[\frac{1}{\Phi(\mathbf{Z}|\mathbf{X})}\mathbb{E}_{q(\mathbf{u})}\left[\frac{1}{q(\mathbf{u})}\int p(\mathbf{y}, \mathbf{X}, \mathbf{f}, \mathbf{u}, \mathbf{Z})d\mathbf{f}\right]\right] \tag{2}$$

Because the joint distribution $p(\mathbf{y}, \mathbf{X}, \mathbf{f}, \mathbf{u}, \mathbf{Z})$ factors as in Equation 1, we can further rewrite the integral of the joint distribution over $\mathbf{f}$ as:

$$\int p(\mathbf{y}, \mathbf{X}, \mathbf{f}, \mathbf{u}, \mathbf{Z})d\mathbf{f} = \mathbb{E}_{p(\mathbf{f}|\mathbf{u}, \mathbf{Z})}\left[p(\mathbf{y} \mid \mathbf{f})p(\mathbf{u})\Gamma(\mathbf{X} \mid \mathbf{Z})p(\mathbf{Z})\right]$$

Plugging this expected value form of the joint distribution integral into Equation 2 and repeatedly applying Jensen's inequality, we derive an evidence lower bound

$$\log p(\mathbf{y}, \mathbf{X}) \geq \mathbb{E}_{p(\mathbf{f}|\mathbf{u},\mathbf{Z})q(\mathbf{u})\Phi(\mathbf{Z}|\mathbf{X})} \left[\log p(\mathbf{y} \mid \mathbf{f})\right] + \mathbb{E}_{\Phi(\mathbf{Z}|\mathbf{X})} \left[\log \Gamma(\mathbf{X} \mid \mathbf{Z})\right]$$
$$- \text{KL}(\Phi(\mathbf{Z} \mid \mathbf{X}) \,||\, p(\mathbf{Z})) - \text{KL}(q(\mathbf{u}) \,||\, p(\mathbf{u})) \tag{3}$$

**The ELBO in practice.**   Rephrasing Equation 3 in terms of the standard SVGP (sparse variational GP; [15]) and VAE ELBOs, we see that:

$$\mathcal{L}_{\text{joint}} = \mathbb{E}_{\Phi(\mathbf{Z}|\mathbf{X})} \left[\mathcal{L}_{\text{SVGP}}(\theta_{\mathcal{GP}}, \theta_{\Phi}; \mathbf{y}, \mathbf{Z})\right] + \mathcal{L}_{\text{VAE}}(\theta_{\Phi}, \theta_{\Gamma}; \mathbf{X})$$

The ELBO can be estimated as follows. Select a minibatch of inputs (e.g., molecules) $\mathbf{X}$, some of which have known labels $\mathbf{y}$, and some of which are unlabeled. Pass the minibatch $\mathbf{X}$ through the encoder $\Phi$ and generate latent codes $\mathbf{Z}$. For latent codes $\mathbf{Z}_\ell$ corresponding to labeled inputs, compute $\mathcal{L}_{\text{SVGP}}$ on the supervised dataset $(\mathbf{Z}_\ell, \mathbf{y}_\ell)$. For *all* latent codes $\mathbf{Z}$, pass through the decoder and add the standard VAE loss. Because the variational posteriors $\Phi(\mathbf{Z} \mid \mathbf{X})$ and $q(\mathbf{u})$ support reparameterization, computing derivatives of the ELBO with respect to $\theta_\Phi, \theta_\Gamma, \theta_{\mathcal{GP}}$ is straightforward. Because the encoder parameters $\theta_\Phi$ are updated through the SVGP loss, $\Phi$ can be viewed as simultaneously acting as an encoder for the VAE and as a (stochastic) deep kernel [52, 51] for the GP model.

**Recentering**   Under this joint model, local BO can still be applied using the same setup and hyperparameters discussed in subsection 3.1. However, when using the above loss, the VAE–and therefore the latent space–is now modified after each iteration of BO. Therefore, the center of the trust region $\hat{\mathbf{z}}$ and latent codes of inputs within or near the trust region may change. To remedy this, we pass observed data $\mathbf{X}$ back through the updated encoder to determine the new location of the corresponding latent points $\mathbf{Z}$ before selecting candidates, and update the trust region accordingly.

**Computational considerations.**   Updating the full encoder $\Phi(\cdot)$ can be significantly more expensive than training the surrogate model only. A practical extension to the above idealized process is to only train in a joint fashion periodically. Thus, the variational GP parameters $\theta_{\mathcal{GP}}$ are updated in each iteration of optimization, but not necessarily $\theta_\Phi, \theta_\Gamma$.

We propose to tie updates to $\theta_\Phi, \theta_\Gamma$ to the sequential successes and failures already tracked by the trust region method. If the optimization procedure is making rapid progress, updating $\theta_\Phi, \theta_\Gamma$ may be an unnecessary expense. Therefore, we update the full set of parameters $\theta_\Phi, \theta_\Gamma, \theta_{\mathcal{GP}}$ after $\tau_{\text{retrain}}$ successive failures. Experimentally we use $\tau_{\text{retrain}} \ll \tau_{\text{fail}}$ as this allows for multiple joint model updates for each time a trust region shrinks.

## 4   Related Work

Bayesian optimization for molecule discovery can be performed without DAEs over fixed lists of molecules [50, 16, 13]. However, the need to pre-define the molecules that can be queried upfront puts a limit to the chemical space such methods can consider: the largest virtual libraries are order of $\approx 10^{10}$ in size [45], which is tiny compared to the space of all possible compounds [26].

Latent-space Bayesian optimization [12, 6, 43, 14, 40, 21] ties the optimization algorithm to a DAE so that, in theory, it can generate any molecule. We broadly group many recent advancements in this approach into two categories: (a) those that investigate new training losses to improve surrogate learning of the objective [14, 43, 6, 5] and (b) those that develop new decoder architectures, for instance improving the validity of the generated molecules by building in grammars or working explicitly in the graph domain [28, 21, 38, 22, 4]. Decoder architectures have also been designed to apply these ideas to different discrete domains such as neural network architectures [55] or Gaussian process kernels [29].

Components of our method including the joint training procedure described in subsection 3.2 are similar to Eissman et al. [6] which considers injecting semisupervision into the VAE via an auxiliary neural network and Grosnit et al. [14] which does so using a triplet loss. In particular, the triplet loss of Grosnit et al. [14] likely encourages similar latent space organization to the method described in subsection 3.2 due to the use of distances in both the triplet loss and the GP covariance function. However, these works use off-the-shelf global Bayesian optimization algorithms, such as expected improvement, as the latent optimizer.

Alternative optimization algorithms have also been proposed for molecule discovery, such as those based on genetic algorithms [20, 34], Markov chain Monte Carlo [53], or ideas from reinforcement learning (RL) [36, 54, 56]. Segler et al. [39] use the cross entropy method to guide the generation of complete SMILES strings from an autoregressive recurrent neural network. You et al. [54] train a policy to build molecules up from atoms and bonds using an RL algorithm. While such methods are often able to find molecules with better property scores than early LS-BO approaches, these methods frequently neglect sample efficiency, impeding their use on practical problems.

## 5  Experiments

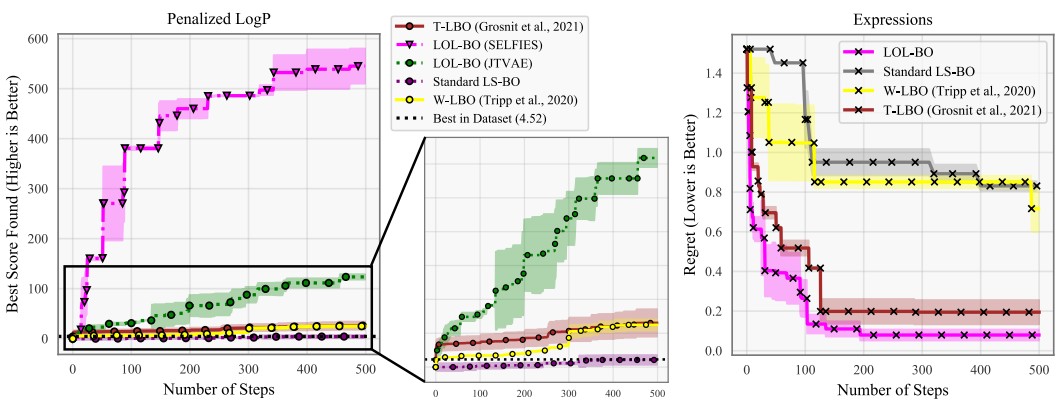

Figure 1: Optimization results for the log P and arithmetic expressions tasks. We compare to recent competitive latent BO methods [14, 43] using the same DAE (`LOL-BO (JTVAE)`) and the new SELFIES VAE (`LOL-BO (SELFIES)`).

We apply `LOL-BO` to six high-dimensional, structured BO tasks over molecules [18, 19] and arithmetic expressions. We additionally investigate the impact of various components of our method in subsection 5.4 and subsection 5.4. log P and Expressions (subsection 5.1) are tasks that are typically evaluated in low budget settings. The Guacamol benchmarks (subsection 5.2) afford larger function evaluation budgets, and the protein docking task (subsection 5.3) is both low budget and the oracle is expensive. Error bars in all plots show standard error over runs. We implement `LOL-BO` leveraging BoTorch [2] and GPyTorch [10][1]. Code and model weights are available at https://github.com/nataliemaus/lolbo.

**Datasets**  All methods have access to the same amount of supervised and unsupervised data for each task. To initialize all molecular optimization tasks, we generate labels for a random subset of $10,000$ molecules from the standardized unlabeled dataset of 1.27M molecules from the Guacamol benchmark software [3]. For the expressions task, we use the same labeled dataset of $40,000$ expressions from [14] to initialize all methods. Furthermore, wherever different VAEs are used for a given task, we use the same unsupervised dataset to train the VAEs so that the amount of unsupervised data also remains consistent. In particular, all molecular VAEs are pre-trained on the same aforementioned dataset of 1.27M molecules from Brown et al. [3].

**Deep autoencoder models.**  A variety of VAE models have been used for the benchmarks tasks we consider below. For `Expressions`, we follow recent work [28, 14, 43, 35] in using a GrammarVAE model [28]. For the remaining *de novo* molecule design tasks, we consider two models.

**Junction Tree VAE (JTVAE).**  Molecules are commonly represented initially using the Simplified Molecular-Input Line-Entry System (SMILES) [49] string representation. SMILES representations are well-known to be brittle, in that single character mutations can easily result in strings that no longer encode a valid molecule. The JTVAE model Jin et al. [21] addresses the problem of generating valid strings by using a molecular graph structure to build each molecule out of valid sub-components so that the generated molecules are almost always valid. As a result, the JTVAE has been widely

---

[1]BoTorch and GPyTorch are released under the MIT license

used as a VAE model for latent space optimization due to its much higher rate of decoding to valid molecules [14, 43, 5, 35]. We follow prior work, using a latent dimensionality of 56.

**SELFIES VAE**   The SELFIES string representation for molecules is a recently proposed alternative to the SMILES string representation [27]. Notably, nearly every permutation of SELFIES tokens encodes a valid molecule. As a result, purely sequence based models using the SELFIES representation are capable of generating valid molecules at a rate competitive with the JT-VAE while avoiding expensive computation involving graph structures.

In our molecule design experiments below, we therefore evaluate a VAE architecture based on the SELFIES string representation in addition to the commonly used JTVAE. We train a VAE model with 6 transformer encoder and transformer decoder layers and a latent space dimensionality of 256 [46]. We use our SELFIES VAE for all molecular optimization tasks below. We also use the JTVAE model to enable direct comparison to prior work.

**Hyperparameters.**   For the trust region dynamics, all hyperparameters including the initial base and minimum trust region lengths $L_{\text{init}}$, $L_{\text{min}}$, and success and failure thresholds $\tau_{\text{succ}}$, $\tau_{\text{fail}}$ are set to the TuRBO defaults as used in Eriksson et al. [8]. Our method introduces only one new hyperparameter, $\tau_{\text{retrain}}$, which we set to 10 in all experiments.

## 5.1   log P and Expressions

In this section, we consider two tasks commonly used in the latent space BO literature–optimizing the penalised water-octanol partition coefficient (log P) over molecules, and generating single variable arithmetic expressions (e.g., $x \times sin(x \times x)$) [14, 43, 5, 35, 28, 1]. Because of the small evaluation budget typically considered for these benchmarks, we use a batch size of 1.

**Baselines.**   For these tasks, we compare to standard latent space BO, as well as a variety of recent prior work in this area. We compare to LS-BO with weighted retraining (`W-LBO`) as described in Tripp et al. [43] and LS-BO with triplet loss retraining (`T-LBO`) from Grosnit et al. [14]. A number of techniques outside of the BO literature consider this task as well. These papers report final objective values $\geq 30$, but do not consider sample efficiency. For example, GEGL [1], which we do compare to in subsection 5.2, uses more than 500 evaluations per iteration. See Ahn et al. [1] for an evaluation on log P.

**Results.**   In Figure 1, we plot the cumulative best objective value for the penalized log P and expressions tasks, averaged over three runs. For log P, `LOL-BO` using the SELFIES transformer VAE achieves an average log P score of 545.17 after 500 iterations. Using a JTVAE, which is more directly comparable to prior work, we still achieve an average score of 123.54–higher than previous state-of-the-art scores using latent space BO of 26.11 [14]. For the expressions task, `LOL-BO` achieves a final regret of 0.079 on average, versus 0.194 achieved by `T-LBO`.

**Molecule quality.**   While molecules found using `LOL-BO` for the log P task are "valid" according to software commonly used to compute these scores, the molecules produced by this search clearly abandon any notion of reality. A number of methods have been proposed to evaluate molecule quality [3], and indeed even some recent work in LS-BO has explored encouraging higher quality solutions [35]. However, our primary goal in this paper is to develop stronger optimization routines in this space, and indeed the baselines we consider also treat log P as an unconstrained optimization problem. Therefore, this result should be taken primarily as evidence that the log P objective can be exploited by a sufficiently strong optimizer, rather than as evidence of novel interesting molecules.

## 5.2   Guacamol Benchmark Tasks

The Guacamol[2] benchmark suite [3] contains scoring oracles for a variety of molecule design tasks, with scores ranging between 0 and 1. We select three tasks among the most challenging in terms of the best objective value achieved: `Perindopril MPO`, `Ranolazine MPO`, and `Zaleplon MPO`. The goal of each of these tasks is to design a molecule with high fingerprint similarity to a known target drug but that differs in a specified target way. The task definitions are discussed in Brown et al. [3].

---

[2]Guacamol is released under the MIT license

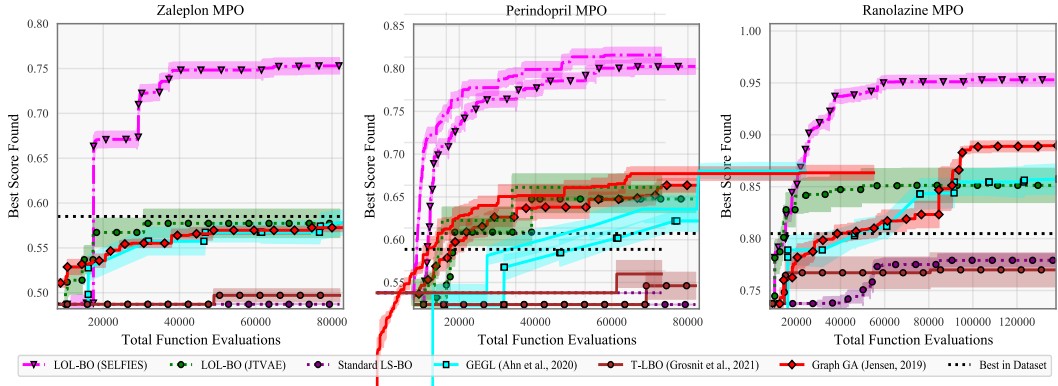

Figure 2: Optimization on three Guacamol benchmark tasks. We compare to T-LBO, the most competitive latent space BO method from above, using the same DAE and SMILES representation (`LOL-BO (JTVAE)`) as well as our proposed SELFIES transformer (`LOL-BO (SELFIES)`). We additionally compare to two recent orthogonal methods for these tasks that do not use deep autoencoders or BO (GraphGA, GEGL). GEGL uses a combination of SMILES representations and directly modifying atomic graph structure, while GraphGA operates entirely on graph structures and only uses string representations for saving and loading molecules.

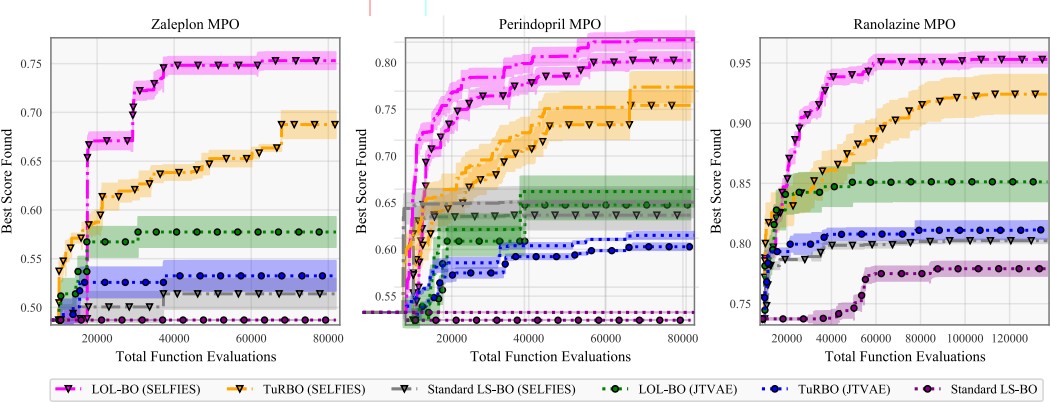

Figure 3: We evaluate (1) the gain in optimization performance achieved in moving from global BO to local (Standard LS-BO vs TuRBO), and (2) the gain in optimization performance achieved adapting the latent space to be suitable for the trust region method (TuRBO vs `LOL-BO`). Improving the optimization algorithm results in significant improvements using either VAE.

**Baselines.**    We compare to `T-LBO`, the most competitive Bayesian optimization method overall from subsection 5.1. Additionally, a variety of techniques from beyond the BO literature using reinforcement learning and genetic programming have been evaluated on these tasks. We compare to GEGL and GraphGA, with GraphGA [20] among the top methods on the Guacamol leaderboard. We also report the best score achieved over the ≈1.27 million molecules in the full Guacamol dataset, which serves as a performance cap for virtual screening (see section 4).

**Results.**    The results of optimization on these three benchmarks are shown in Figure 2. We again report results for `LOL-BO` with both a JTVAE and SELFIES VAE deep autoencoder. `LOL-BO` with a SELFIES VAE optimizes significantly faster than other methods in the evaluation budget used here.

### 5.3   DRD3 Receptor Docking Affinity

The Therapeutics Data Commons (TDC) benchmark suite [17] contains oracles that evaluate the binding propensity of a ligand (small molecule) to a target protein. These docking oracles incur nontrivial computation cost, requiring on the order of minutes to evaluate, making them ideal

Table 1: **(Left.)** Comparing SELFIES VAE and JTVAE on metrics computed over molecules in the GuacaMol test dataset [3]. Decode stddev is the standard deviation in `Perindopril MPO` objective value when decoding a fixed latent $\mathbf{z}$ 20 times, averaged over a subset of 3500 latent vectors from the dataset. **(Right.)** Docking scores achieved by various methods after the four evaluation budgets considered on the TDC leaderboard (lower is better). We compare to three BO baselines and a variety of orthogonal approaches from the TDC leaderboard.

| | SELFIES VAE | JTVAE |
|---|---|---|
| Reconstruction Acc. | **0.913** | 0.767 |
| Avg. Decode Time (s) | **0.02** | 3.14 |
| Decode Stddev | **0.04** | 0.1 |
| Validity | 1.0 | 1.0 |
| GP Test NLL (Perindopril) | **-2.156** | -1.481 |
| GP Test NLL(Zaleplon) | **-2.705** | -1.781 |
| GP Test NLL (Ranolazine) | **-2.127** | -1.649 |

| Method | Best Score Found in Evaluation Budget | | | |
|---|---|---|---|---|
| | 100 evals | 500 evals | 1000 evals | 5000 evals |
| LOL-BO (SELFIES) | $-\mathbf{13.1 \pm 0.3}$ | $-\mathbf{13.3 \pm 0.1}$ | $-\mathbf{13.8 \pm 0.5}$ | $-15.4 \pm 1.3$ |
| TuRBO (SELFIES) | $-12.6 \pm 0.0$ | $-12.7 \pm 0.3$ | $-12.9 \pm 0.4$ | $-13.0 \pm 0.4$ |
| LS-BO (SELFIES) | $-12.6 \pm 0.0$ | $-12.6 \pm 0.0$ | $-12.6 \pm 0.0$ | $-12.6 \pm 0.0$ |
| LOL-BO (JTVAE) | $-12.6 \pm 0.0$ | $-12.7 \pm 0.1$ | $-12.8 \pm 0.2$ | $-13.1 \pm 0.3$ |
| TuRBO (JTVAE) | $-12.6 \pm 0.0$ | $-12.6 \pm 0.0$ | $-12.7 \pm 0.3$ | $-12.7 \pm 0.3$ |
| LS-BO (JTVAE) | $-12.6 \pm 0.0$ | $-12.6 \pm 0.0$ | $-12.6 \pm 0.0$ | $-12.6 \pm 0.0$ |
| T-LBO | $-12.6 \pm 0.0$ | $-12.6 \pm 0.0$ | $-12.6 \pm 0.0$ | $-12.8 \pm 0.2$ |
| Graph-GA | $-11.8 \pm 1.1$ | $-12.5 \pm 0.7$ | $-13.2 \pm 0.7$ | $-\mathbf{16.5 \pm 0.3}$ |
| SMI-LSTM | $-11.1 \pm 0.6$ | $-11.4 \pm 0.6$ | $-12.0 \pm 0.2$ | $-14.5 \pm 0.5$ |
| MARS | $-9.1 \pm 0.7$ | $-9.8 \pm 0.3$ | $-11.1 \pm 0.1$ | $-11.4 \pm 0.5$ |
| MolDQN | $-7.0 \pm 0.2$ | $-7.6 \pm 0.2$ | $-7.8 \pm 0.04$ | $-10.0 \pm 0.2$ |
| GCPN | $-11.6 \pm 2.2$ | $-12.0 \pm 0.7$ | $-12.0 \pm 0.6$ | $-12.3 \pm 1.0$ |

benchmark tasks for Bayesian optimization. In this paper, we focus on designing ligands that bind to dopamine receptor D3 (DRD3), as the TDC website maintains a leaderboard of results for DRD3 [3]. We compare LOL-BO on the DRD3 docking task to the TDC leaderboard and three BO methods–T-LBO, standard LS-BO, and fixed latent space TuRBO on this task.

Results of optimization are in Table 1 (right). LOL-BO achieves the best performance among the BO baselines, with only TuRBO making significant progress on the task by 5000 evaluations. Compared to the baselines from the TDC leaderboard, LOL-BO is significantly more sample efficient, achieving results comparable to the best method with a factor of $10\times$ fewer oracle calls.

### 5.4 Ablation studies

In this section, we evaluate the components of LOL-BO described in subsection 3.2 beyond the direct application of TuRBO in the latent space described in subsection 3.1. We run TuRBO on the three Guacamol tasks considered in subsection 5.2 using a fixed latent space extracted by the pretrained SELFIES VAE and JTVAE. To run TuRBO, we use the same experimental setup as for LOL-BO, but $\Phi(\cdot)$ and $\Gamma(\cdot)$ are fixed at the start of optimization. We compare LS-BO, TuRBO, and LOL-BO on all three tasks in Figure 3. The use of SVGP is controlled across methods.

LOL-BO outperforms TuRBO across all DAE models and tasks. While these results demonstrate the impact of joint training, the excellent performance of TuRBO in a static, pre-trained latent space highlights the value of high dimensional Bayesopt in this setting.

To further analyze whether the jointly trained latent space better-matches the distance based assumptions of the GP model, we include an additional analysis directly comparing scores of molecules sampled from a local region around the top-scoring molecule in the latent space of the pretrained SELFIES-VAE before optimization has begun and at the end of optimization (see subsection A.3).

Since LOL-BO achieves better performance with the SELFIES VAE than with the JTVAE, we evaluate a series of metrics for both DAE models. In Table 1 left, we compare reconstruction error and validity, as well as the standard deviation in `Perindopril MPO` score from decoding a fixed latent vector $\mathbf{z}$ 20 times, averaged over 3500 distinct $\mathbf{z}$. We report supervised learning performance for a GP trained on $10,000$ molecules taken from the initial Guacamol dataset, using the *pretrained* latent spaces. Across standard metrics, the SELFIES VAE achieves excellent performance. Beyond these, the improvement in objective variance for a single $\mathbf{z}$ is notable: because Guacamol objective values are in the range $[0, 1]$, a standard deviation of $0.1$ represents significant additional noise. Thus, the DAE can be a significant source of noise even for an otherwise deterministic objective function.

---

[3] https://tdcommons.ai/benchmark/docking_group/drd3/

# 6 Conclusion

Designing latent spaces for Bayesian optimization is challenging because the VAE and the optimizer often compete: too small of a latent space renders the VAE incapable of capturing the structure present in the input space, while too large a latent space makes optimization challenging.

The empirical results we present here demonstrate the importance of viewing these latent spaces as high-dimensional and tackling optimization with high-dimensional BO strategies, even though these latent spaces are often significantly smaller than the input space. Furthermore, we show that high-dimensional BO methods can be significantly improved when translating them to the latent space setting, demonstrating the potential of further research at the intersection of these areas.

# 7 Acknowledgements

JRG and NM were supported by NSF award IIS-2145644. JM and HJ were supported by the Laboratory Directed Research and Development program of Los Alamos National Laboratory (LANL) under project number 20210043DR.

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
