# A Appendix

## A.1 Additional implementation details

As the data collected by `LOL-BO` grows, it becomes increasingly computationally expensive to update the models in and end-to-end fashion on all of the data. Thus, instead of updating on all data, we only update end-to-end on a subset of the data consisting of the most recently collected batch of data points, plus the top-$k$ scoring points from the remaining data. For the experiments in subsection 5.1 and subsection 5.3 where we have a small evaluation budget, we use $k = 10$. For the experiments in subsection 5.2 with a larger total evaluation budget, we use $k = 1000$.

## A.2 JTVAE vs. SELFIES VAE additional analysis

In Figure 4, we provide histograms to visualize a few of the statistics summarized in Table 1 in the main text 1.

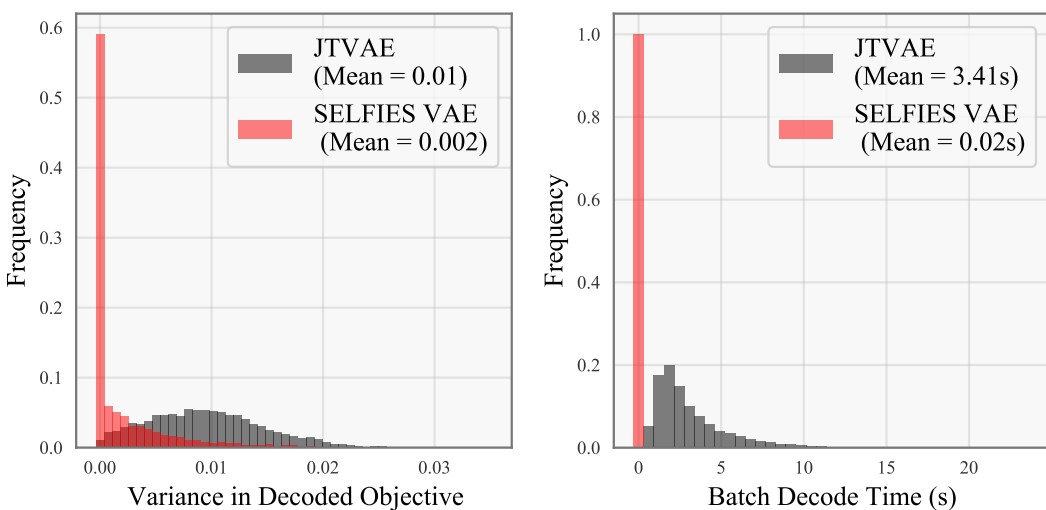

Figure 4: (*Left*) Histogram of variance in Perindopril MPO objective value across 20 molecules decoded from a single a latent code **z**. (*Right*) Time taken to decode a batch of points.

## A.3 Additional ablation analysis

In this section, we analyze whether the jointly trained latent space matches the distance based assumptions of the GP model. On the Ranolazine MPO task, we first encode the best molecule $x^*$ (score=0.9382) from a single run of `LOL-BO` to get the latent codes $z^*_{\text{pre}}$ and $z^*_{\text{joint}}$ from both the pretrained VAE before optimization has begun and at the end of optimization. We then analyze the average score of molecules decoded from latent vectors sampled uniformly from an $L$-hypercube centered at both $z^*_{\text{joint}}$ and $z^*_{\text{pre}}$. By $L = 5$, latent codes in the joint VAE near $z^*_{\text{joint}}$ still achieve an average score of 0.9106, while for vectors near $z^*_{\text{pre}}$ in the pretrained model the average score has dropped to only 0.3382 Figure 5.

In this section, we additionally analyze whether the jointly trained VAE is better able to reconstruct the highest-scoring molecules found by the optimizer. Empirically, the optimizer consistently produces higher-scoring molecules than any found in the entire initial unsupervised dataset used to train the initial VAE (see Best in Dataset comparison in Figure 2). This raises the question of whether these high-scoring molecules found by the optimizer may often be out-of-sample for the initial VAE. To investigate this, we took the top 128 highest-scoring molecules found by `LOL-BO` on the Ranolazine MPO task and computed the reconstruction accuracy of the initial VAE on these molecules. We repeated this experiment with the final VAE obtained at the end of the optimization run. We found that the average reconstruction accuracy of the initial VAE was 0.01 while the average accuracy of the final VAE was 0.89. The extremely low reconstruction accuracy of the initial VAE indicates that many of the high-scoring molecules found by the optimizer are indeed out-of-sample. It would therefore be very difficult to find these high-scoring molecules by simply searching in the latent space of the initial VAE. In contrast, finding these high-scoring molecules by searching in the latent space of the final VAE would likely be much easier. Therefore, this result provides further motivation for our procedure of jointly updating the VAE during optimization, so that as the optimization proceeds, the VAE is made to adapt so that it can better construct the types of moleucles we are interested in, even if they are unlike any molecules in the initial unsupervised dataset.

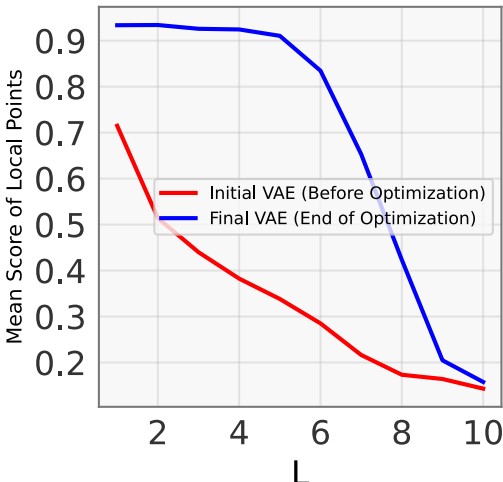

Figure 5: Comparing the latent spaces of the SELFIES-VAE before and after running `LOL-BO` to optimize Ranolaize MPO. Plot shows mean Ranolaize MPO score of 100 decoded latent points randomly selected from within a hypercube of length $L$ centered on the top scoring molecule found during optimization.

## A.4   Compute

The SELFIES VAE was trained using a single server with eight Quadro RTX 8000 GPUs. All optimization experiments were run on a single RTX A5000 GPU. We used an internal cluster for compute.

## A.5   Software licensing

All major software packages used (ie BoTorch, GPyTorch, Guacamol) are released under MIT license.

## A.6   Run-time considerations

`LOL-BO` scales to large evaluation settings readily via minibatch training of the ELBO in Equation 4 due to the variational GP approximation used (Hensman et al., 2013). `TuRBO` scales similarly by using the same variational GP approximation. Although we've shown that `LOL-BO` significantly improves performance over `TuRBO` (see Figure 5), it is important to consider the additional time complexity of `LOL-BO`. In particular, updating the VAE jointly can add significant time complexity, particularly with very deep transformers. However, this cost is generally manageable since relatively little VAE updating is required with the small amount of data acquired at each step (compared to the initial training on the large unsupervised dataset).

When the evaluation budget is small (Section 5.1), or the objective function expensive (Section 5.3), `LOL-BO` incurs as little as a $1.01\times$ slowdown over running `TuRBO`. However, larger evaluation budgets require retraining the encoder on growing amounts of data. Thus, in Section 5.2, we do incur as much as a $5\times$ slowdown in the worst case. However, the total running time in Section 5.2 is still typically on the order of half a day, and we believe the improved outer-loop optimization performance makes the additional time complexity worthwhile. Finally, the added cost of jointly updating the VAE is negligible compared to the cost of certain oracle calls typical in molecule design. For instance, in Section 5.3 the docking oracle takes minutes to evaluate per molecule, leading to run-times of up to a few days, and even this would be considered cheap compared to wet-lab experiments. To see specific wall-clock times for all methods, see 2. The wall clock times given for W-LBO [43] were obtained with our modifications to parallelize the W-LBO code-base.

## A.7   Potential limitations and future work

This work focuses on molecular optimization tasks centered around drug development. While the development of new drugs has the potential to help many people, it is also important to consider the dual use nature of such methods. In particular, AI technologies for drug discovery have the potential to be misused for the de novo design of biochemical weapons [44]. It is also important to consider the potential for negative societal impact if any new drugs discovered are not property tested before being administered. While we hope that our work can eventually speed-up the drug development process by providing useful candidate molecules, it is essential that

Table 2: Wall clock runtimes for all methods on all tasks.

| Method | Avg. Wall Clock Runtime (hours) | | | |
|---|---|---|---|---|
| | Penalized Log P (Sec. 5.1) | Expressions (Sec. 5.1) | GuacaMol (Sec. 5.2) | Docking (Sec. 5.3) |
| LOL-BO (SELFIES) | 3.0 | NA | 15.4 | 118.4 |
| TuRBO (SELFIES) | 2.8 | NA | 4.2 | 104.3 |
| LS-BO (SELFIES) | 2.8 | NA | 4.1 | 103.8 |
| LOL-BO (JTVAE) | 8.9 | NA | 49.1 | 132.7 |
| TuRBO (JTVAE) | 7.5 | NA | 15.8 | 116.9 |
| LS-BO (JTVAE) | 7.5 | NA | 15.6 | 117.5 |
| LOL-BO (Grammar VAE) | NA | 0.83 | NA | NA |
| TuRBO (Grammar VAE) | NA | 0.78 | NA | NA |
| LS-BO (Grammar VAE) | NA | 0.78 | NA | NA |
| T-LBO | 30.0 | 1.5 | 107.2 | 138.0 |
| W-LBO | 10.0 | 5.0 | NA | NA |
| GEGL | NA | NA | 1.3 | NA |
| Graph-GA | NA | NA | 0.17 | NA |

experts remain heavily involved and that FDA guidelines for drug development and approval are strictly adhered to.

Additionally, the oracles and tasks we use here, even ones with the potential to design candidate drug molecules like protein docking, are completely oblivious to critical aspects like human safety and efficacy, stability, manufacturing cost, and so on. These computational approaches to molecule design should therefore be considered first pass screening procedures, not procedures to design final end products.

In order to optimize the Guacamol molecular optimization tasks, LOL-BO requires tens of thousands of function evaluations Figure 2. While the fairly cheap Guacamol oracles have made it reasonable to do this many evaluations, there are many molecule design applications where each function evaluation is much more expensive. In particular, tens of thousands of function evaluations is likely not efficient enough for many applications that would be run in vitro with actual wet-lab experiments.

Another potential limitation of LOL-BO is that is is designed to find a single best optima for a given task. However, in the molecular space in particular, it is often desirable to find multiple good optima since a single solution might fail for unknown reasons downstream. In future work, we plan to explore methods that can be used to extend Bayesian optimization to obtain a set of diverse solutions rather than a single solution.