# OpenReview forum: "Local Latent Space Bayesian Optimization over Structured Inputs"
_NeurIPS.cc/2022/Conference — NeurIPS 2022 Accept_

### Official Review · Reviewer_B8Dz · 2022-07-09

**Rating:** 6
**Confidence:** 4
**Soundness:** 3 good
**Presentation:** 3 good
**Contribution:** 2 fair

**Summary:**

The paper deals with solving the high dimensional Bayesian optimization problem with latent space deep autoencoder models (DAE) primarily focusing on structured input. The idea revolves around the high dimensionality issues that persist even after projecting the structured input into the latent with DAE models. The primary novelty lies in efficiently incorporating a trust region in the Bayesian optimization step to deal with this high-dimensionality issue, not explicitly addressed in the earlier research. The empirical performance demonstrates significant improvements over the latest SOTA latent space Bayesian optimization algorithms across various complex real-world benchmarks. The paper tries to emphasize improving the optimization methodologies as well along with the improvement of Deep generative models and shows its effectiveness for latent space BO with a combination of both.


**Questions:**

1. What was the initial sample size for both the unsupervised and supervised cases? Can a comparison be provided with the other methods as well? I am specifically interested to see when the initial samples especially supervised information less how the performance compares?

2. Joint-inferencing and using Transformer architectures will definitely require more initial sets of samples right? Unsupervised information can be available but there should be a proper justification for that. Hence, I am curious to understand the initial information comparison.

3. TuRBO maintains m trust regions simultaneously. It uses a TS type approach to choose the best sample across the trust regions which can be paralleled and results in better efficiency. However, this work doesn't consider multiple trust regions, but still, there is an improvement over TurBo. Is it due to the Sparse GP Augmentation aspect with a global set of learned-inducing variables?

**Limitations:**

The authors have addressed the computation challenges to some extent and have carefully addressed the societal challenges which are appreciable. They clearly mention that experts should be heavily involved and that FDA guidelines for drug development and approval should be strictly followed.

**Strengths And Weaknesses:**

The main strength revolves around addressing the mismatch b/w the trust-region in the latent space and  trust-region in the original structured input space. Although an extremely efficient formulation of trust region-based local optimization has been proposed by Eriksson et. al [2] that doesn't deal with the high-dimensionality with latent space BO. One of the most important contributions of this paper lies in formulating the trust-region for a high dimensional latent space Bayesian optimization problem. This paper points out a very interesting and critical point that even a small change in the latent space generated by BO i.e $||z - z'|| \leq \epsilon$ can be drastically different in the original space since the closeness in DAE is defined w.r.t to the decoder and BO doesn't explicitly incorporates that, hence it can still be that $||x - x'|| \geq \Delta$ which contradicts the purpose of trust-region based BO objective as pointed in Eriksson et. al. The main strength of this paper, according to me lies in the joint inference formulation which allows simultaneous optimization of GP with VAE parameters. The paper proposes an interesting methodology combining several popular LSBO/ Trust-region BO methods with State of the art DAE models in an intelligent framework. The empirical evaluations are quite explicit and detailed. In Figure 1, they show that LOL-BO (SELFIES) outperforms drastically all other benchmarks which are interesting. But what is even more critical is that the LOL-BO (JT-VAE) outperforms other baselines as well.

The primary weakness of this research lies in its uniqueness and novelty since it seemed like combining the best aspects of the latest advancements in the field of LSBO, Trust-region BO, and High dimensional BO in general. For ex: The semi-supervised variational inference was introduced by Kingma et. al and has been used to replace the weighted re-training mechanism popular in the LSBO literature by Miguel et al. Similarly, the paper by Eriksson et. al had already introduced the notion of trust-region in Bayesian optimization but it didn't have the latent space aspect to it. Also, the authors incorporate transformer layers on the recently published SELFIES representation which shows significant improvements. So, it's not extremely clear the exact novelty of this research. Another challenge lies in jointly learning multiple parameters and also using deep generative models should require a larger initial dataset both supervised and unsupervised and a comparison of that is not explicitly given as the major purpose of BO and GP, in general, is to be effective in a low-data regime where the initial supervised information is less. Hence, a comparison is missing with the initial supervised (and unsupervised) data on which the GP is trained? Also, related to this lies the concern in the simultaneous updating of the parameters, which the authors have dealt with various heuristics of periodically training, etc. However, there are no guarantees of convergence provided for the same.

### Post Rebuttal
I am happy with the author's response and increased my score.


References :

[1]. Austin Tripp, Erik A. Daxberger, and José Miguel Hernández-Lobato. Sample-efficient optimization in the latent space of deep generative models via weighted retraining. In Advances in Neural Information Processing Systems 33, 2020

[2]. David Eriksson, Michael Pearce, Jacob Gardner, Ryan D Turner, and Matthias Poloczek. Scalable global optimization via local Bayesian optimization. In H. Wallach, H. Larochelle, A. Beygelzimer,  F. d'Alché-Buc, E. Fox, and R. Garnett, editors, Advances in Neural Information Processing Systems, volume 32. Curran Associates, Inc., 2019. URL https://proceedings.neurips.cc/paper/2019/file/6c990b7aca7bc7058f5e98ea909e924b-Paper.pdf

[3]. Antoine Grosnit, Rasul Tutunov, Alexandre Max Maraval, Ryan-Rhys Griffiths, Alexander Imani Cowen370 Rivers, Lin Yang, Lin Zhu, Wenlong Lyu, Zhitang Chen, Jun Wang, Jan Peters, and Haitham Bou-Ammar. High-dimensional Bayesian optimization with variational autoencoders and deep metric learning. CoRR, abs/2106.03609, 2021. URL https://arxiv.org/abs/2106.03609

[4]. Diederik P Kingma, Shakir Mohamed, Danilo Jimenez Rezende, and Max Welling. Semi-supervised learning with deep generative models. In Advances in neural information processing systems, pages 3581–3589, 2014

---

> ### Author Response · Authors · 2022-08-01
> **Response to questions and comments**
>
> Thank you for your time in reviewing our work and your positive feedback. We especially agree with you that a more thorough discussion at the start of Section 5 of the datasets used is warranted, particularly to clarify and emphasize that all methods have access to the same amount of supervised and unsupervised data, more details on this below. We will add this discussion to the main text.
>
> **(Question 1) Comparison between supervised and unsupervised data:** Critically, we use the same small random supervised dataset to initialize optimization as well as the same set of unlabeled data for all methods. For unlabeled data, all methods use the standardized unlabeled dataset of 1.27M molecules provided by the Guacamol benchmark software for their tasks. Further, all methods have seen identical quantities of labeled data at any point along the x-axis across all figures. Thank you for mentioning this, we will make it clearer that the x-axis in all figures is the total number of labels available to a given method at that point, both from initial data and data collected during optimization.
>
> **(Question 2):** As mentioned above, neither the semi-supervised joint training of LOL-BO nor the transformer model used additional supervised or unsupervised data beyond what is used by other methods. It is indeed possible as you suggest that the transformer model specifically would benefit from being trained on even more unsupervised data (e.g., perhaps the full ZINC20 dataset), as transformer models may continue to improve with even larger datasets–we plan to evaluate this in future work.
>
> **(Question 3) Considering more than 1 trust region:** Good question. As you mention in TuRBO, the number of trust regions m can be changed, and in the original paper (Eriksson et al., 2020) they find that on many tasks (in the paper, all except Cosmo and possibly Lunar in Figure 3), m=1 is often competitive or even better than m>1. In the tasks here, we similarly simply found that m>1 generally provided no benefit, but introduced additional complexity. Thus all results for both LOL-BO and TuRBO are with m=1 (e.g., “TuRBO” should be taken to mean TuRBO-1 from Eriksson et al., 2020). On your second question about the improvement over TuRBO, this is due to the end-to-end retraining of the latent-space used by LOL-BO to adapt it to the optimization task – the use of SVGP is controlled for in the Guacamol tasks, as both TuRBO and LOL-BO must use an SVGP model to scale to the large evaluation budgets. Thanks again for your question, and we will clarify these points in the main text.
>
> **On convergence:** Thanks for this question. On iterations when end-to-end retraining happens, we make sure to let the joint GP and VAE training converge, i.e., let gradient descent on the ELBO of the joint model converge to a critical point. We agree that it would be difficult to analyze convergence of the joint model to a stationary point over the course of the optimization. There may even be evidence against this: empirically (c.f. Figure 2), the optimizer consistently produces better scoring molecules than those found in the entire initial unsupervised dataset, and thus produces molecules that are out of sample for the VAE and require retraining updates. From this perspective, we believe this actually motivates the end-to-end updates, as the VAE needs to be updated on molecules produced by the optimizer to be able to reliably reconstruct them. To add evidence for this motivation, we will include an additional experiment in the appendix that demonstrates that the initial VAE has significantly lower reconstruction accuracy on molecules discovered by the end of optimization. Thank you for bringing this up!

---

> > ### Comment · Reviewer_B8Dz · 2022-08-07
> > **Additional comments and concerns**
> >
> > Thanks for the detailed comments and were really helpful. I appreciate the comments where the authors clearly highlight the shortcomings and address them for future work.
> >
> > However, I am still not completely convinced about the fact that this approach can work when the initial supervision set is small and if yes, can the authors please provide an additional justification for the same? My primary point of concern is that one of the major aspects of Bayesian Optimization is when the initial supervision set is small, especially for various critical decision-making problems. In such scenarios, why it should perform better than JT-VAE (VAE) is still not clear.
> >
> > I liked the discussion around Question 3 and I believe the author's justification is correct and intuitive. Finally, I thank the author for addressing the convergence issue.
> >
> > One additional point that the authors missed is a comment on the novelty of the research as I felt this as a combination of 3 state of the art methods in High-D BO. Please can the authors throw some light on what are the aspects that differentiate their method significantly? Thanks.

---

> > > ### Author Response · Authors · 2022-08-07
> > > **Initial supervision**
> > >
> > > Thank you for following up! We attempt to further address your concerns below -- please do let us know if you have any remaining questions or concerns.
> > >
> > > **Regarding the initial supervised set:** The empirical results in our paper provide evidence that our approach works better on a variety of data size regimes, even with the SELFIES transformer despite no additional supervised data over other methods. We believe these results provide our best non-speculative argument that we do not need more initial data than other methods, because we achieved better performance despite not using more initial data on any task. On logP and Expressions we collect only 500 labeled examples total, and we collect 5000 examples total on the docking task. Notably on the docking task, TuRBO (SELFIES) alone makes virtually no progress for those 5000 evaluations, while LOL-BO (SELFIES) does, demonstrating that our joint training procedure can be used in the data efficient regime. Finally, while we certainly agree that the extremely data efficient optimization setting has classically been an important application regime for Bayesian optimization, surely BO should not be restricted as a research area to this setting, and we believe BO can be a valuable technology for general high dimensional global optimization problems, including at the budget of the Guacamol tasks.
> > >
> > > In summary, we currently interpret your concern to be that our method may not work with only a small amount of initial supervised data, e.g. due to the transformer and weight updating. However, throughout our paper, we do not use more data than other methods, so the concern is simply not empirically supported on the tasks we consider. It is certainly possible that even smaller budgets would require smaller neural networks (note that the expressions task already uses a much smaller neural network from Kusner et al., 2017) and gain little benefit from joint training. However, the log P and expressions tasks are already the smallest budget tasks we're aware of in the LSBO literature, where our method is empirically beneficial. Beyond empirical results, we think the performance of the transformer in these regimes is not unreasonable or contrary to prevailing wisdom in the literature. Pretraining these models on very large amounts of unsupervised data and then fine tuning them to supervised tasks with much less labeled data is generally known to work well in practice, and our paper simply supports this trend.
> > >
> > > **Regarding novelty:** Ultimately, we believe the simplicity of our method is outweighed by the benefits, namely (1) dramatic performance improvements of up to 20x for LSBO over any existing published method, and (2) a simple platform from which to continue advancing state of the art performance for BO based molecule optimization. As the other reviewers point out, our paper "considers an important real-world application with an effective solution that outperforms multiple baselines" and "unlocks substantial improvements in Bayesian optimization on high dimensional spaces such as molecule graphs".
> > >
> > > We additionally do believe there is some methodological novelty in our paper. The semi-supervised VAE approaches that exist in the literature tend to either use an auxiliary neural network for injecting supervision (e.g., as in the original SS-VAE of Kingma et al) before training a GP on the resulting latent space, or using e.g. the triplet loss of Grosnit et al (your [3], which we compare to). W-LBO (your [1]) only utilizes weighted unsupervised signal obtained during BO to update the transformer. Our extension to these ideas of performing joint variational inference of the VAE and the GP directly is fairly dramatically better than e.g. TuRBO alone in some cases. TuRBO is consistently outperformed by LOL-BO, and makes virtually no progress at all on the docking task. On another Guacamol task, Valsartan SMARTS, LOL-BO not only achieves a nearly perfect score >0.98, but is the only LSBO method we consider in this paper that makes any progress at all and achieves a score above 0, including e.g. TuRBO (SELFIES)--we will add these results. Finally, the SELFIES transformer model carried with it all the standard engineering challenges of getting such a large model to work as well as we could, and we intend to release pretrained models, which we think will themselves have value to the LSBO community as faster, more accurate drop in replacements to the commonly used JT-VAE.

---

> > > > ### Comment · Reviewer_B8Dz · 2022-08-08
> > > > **Final comments & claims**
> > > >
> > > > Thanks for the detailed comments which help in understanding several aspects of the paper.
> > > > I agree with the authors regarding the applicability of BO to more general settings and the overall objective of the paper.
> > > > Although I still believe novelty is a concern, however, I agree and appreciate the simplicity in the formulation and the vast empirical improvements.
> > > > Also, convergence analysis of the work will be critical and encourage the authors to highlight them in the final version.
> > > > I am happy with the response and will improve the scores. Thanks

---

### Official Review · Reviewer_rFLV · 2022-07-10

**Rating:** 7
**Confidence:** 4
**Soundness:** 3 good
**Presentation:** 3 good
**Contribution:** 3 good

**Summary:**

The paper considers the problem of optimizing black-box functions defined over structured inputs. The solution is situated within the general framework of Bayesian optimization (BO) over latent space where a deep-autoencoder model is learned from unsupervised data and subsequently used for employing standard Bayesian optimization algorithms in the continuous (latent) embedding. One challenge with this general approach is the high-dimensionality of the latent space which converts a black-box optimization problem over combinatorial space to that over high-dimensional continuous space. The paper proposes a trust region based BO algorithm in the latent space as a way to address this challenge. Based on the hypothesis that there is a mismatch between the trust region on the latent space and the corresponding trust region in the original structured space, a joint inference procedure is proposed over the Gaussian Process (GP) surrogate model and the deep autoencoder model i.e., treating the deep autoencoder as sort of a deep kernel for the GP model. Experiments are performed on multiple molecular optimization and arithmetic expression tasks.

**Questions:**

Please see above.

**Limitations:**

There is a good description of limitations in the supplementary section.

**Strengths And Weaknesses:**

- The considered problem of black-box functions defined over structured inputs is very significant with many real world applications including molecular design/drug discovery.

- The idea of trust region Bayesian optimization for high-dimensional spaces is well known now especially with the work of Eriksson et al (TurBO)[1]. Therefore, the novelty of the proposed method is relatively limited but it does improve over the standard application of TurBO in latent space.

- The paper presents empirical analysis for the proposed approach. I found the empirical analysis sound and effective in showing the efficacy of the approach on the overall BO performance. The proposed approach outperforms other baselines on multiple benchmarks and it is especially refreshing to see the discussion around usefulness of objectives like log-p metric in molecular optimization (line 308 to 321).

- I found the literature review to be comprehensive and representative of the field.

In summary, the paper considers an important real-world application with an effective solution that outperforms multiple baselines. Barring some time complexity issues with the proposed inference procedure, it should be a useful method for practical applications.

---

> ### Author Response · Authors · 2022-08-01
> **Response to comments on time complexity**
>
> Thank you for your encouraging comments and careful feedback.
>
> We agree that the time complexity of our method could be an important consideration in some situations, and to illustrate this more scientifically, we will add absolute wall clock timing results for all methods to the appendix by extending Appendix A.5, which currently contains relative slowdowns. To summarize, updating the VAE jointly does indeed add time complexity, particularly with very deep transformers. However, this cost is generally manageable since relatively little updating is required with the small amount of data acquired by each update (compared to the initial training on the large unsupervised dataset). Moreover, we believe the improved outer-loop optimization performance makes the additional joint VAE training worthwhile in many molecular optimization tasks, where extremely expensive oracles, like those used for docking, are common. Thank you again for bringing this up though and we will make this more explicit.

---

> > ### Comment · Reviewer_rFLV · 2022-08-07
> > **Final response**
> >
> > Thank you for your reply. I am happy with the response. I think the paper will bring good value to the community.

---

### Official Review · Reviewer_RDPg · 2022-07-12

**Rating:** 7
**Confidence:** 4
**Soundness:** 4 excellent
**Presentation:** 4 excellent
**Contribution:** 3 good

**Summary:**

The paper proposes a fused autoencoder/VGP approach which unlocks substantial improvements in Bayesian optimization on high dimensional spaces such as molecule graphs.

**Questions:**

Equation 2 has identical numerator/denominator terms with $q(u)$ and $\Phi(Z|X)$, which seems incorrect for the variational approximation? (At a glance, should it be p/q and $\frac{\Gamma}{\Phi}$)?

Why not compare SELFIES vs JTVAE in terms of score-achieved-by-iterations, as done w/ TuRBO vs LOL-BO in Table 1 rhs? Adding a row for either TuRBO-JTVAE or LOL-BO-JTVAE to Table 1 right would be helpful to set a clearer baseline.



**Limitations:**

Bayesopt on biologics might be a space where a societal impact statement might be in order. De novo biologicals design would constitute a mixed-use technology.

**Strengths And Weaknesses:**

Originality
The paper brings together a few lines of work: Most importantly, it fuses trust region bayesopt (TuRBO)+sparse/variational GP with learned index point representations (via VAE), reconciling the trainable representation learning with the trust region approach by jointly training with a semisupervised objective. Secondly, the paper incorporates representation improvements (JTVAE=>SELFIES) which provide substantial posterior/eval speedups and higher chances of emitting valid representations in the discrete molecule space. That said, because "all it does" is bring these lines of work together, this work is more about unlocking performance as opposed to proposing a radically original idea -- it's somewhat incremental to TuRBO, to VGP, to SELFIES.

Quality
Well written, problem well framed. Experiments against multiple Guacamol targets + DRD3 as well as Expressions dataset go beyond toy problem to address interesting industry benchmarks. Ablation study is helpful for disentangling the contributions of joint training (LOL-BO) vs representation change (SELFIES). It is evident that both are contributing substantially. It would be nice to have "best score by iterations" type metrics for JTVAE vs SELFIES. Is this an issue of computation budget (100x more expensive JTVAE prohibitive for bayesopt)?.
L310 spelling SELFIS

Clarity
Problem space, experiments are well explained. Joint variational training approach is well explained. I wonder if there is an error in Eq 2 (see questions below).

Significance
The topic area (bayesopt on high dim discrete spaces) is timely and important, and one which the field has not fully solved to-date.

---

> ### Author Response · Authors · 2022-08-01
> **Response to questions and comments**
>
> Thank you for your feedback. We respond to your comments and indicate how we will be adjusting our paper accordingly below.
>
> **(Question 1) Concern about equation 2:** Sorry about the confusion here. The reason for the identical numerator/denominator terms is due to the, potentially overly-verbose, first line of the derivation of the ELBO, where we introduce the log evidence as $ \log p(x) = \log \int p(x, z)dz = \log \int p(x,z)\frac{q(z)}{q(z)} dz $. This is in order to rewrite the log evidence as an expectation over the newly introduced variational distribution q(z). For brevity, the equations following this omit the next step in the derivation, which is the expression we believe you’re alluding to in your review. This would indeed be as you point out: an expectation over q of p/q – we’re happy to include that step between the first and second lines of equation 2 as well in the derivation for improved clarity.
>
> **(Question 2) Suggestion to add "best score by iterations" metrics:** We agree that adding the JT-VAE vs SELFIES breakdown to Table 1 would complete the set of results in the paper, and we’ll add this. As you guessed we found bayesopt with the JT-VAE to be extremely computationally intensive, and thus had initially stopped considering it after the performance benefits of the transformer became clear in the first set of tasks in Figures 1, 2, and 3, which isolate the difference both with joint retraining–LOL-BO (SELFIES) vs LOL-BO (JT-VAE)–and with just the pretrained model alone–TuRBO/LS-BO (SELFIES) vs TuRBO/LS-BO (JT-VAE). But we agree that adding this result would make this point clearer. Thanks for this suggestion. Also thank you for catching the misspelling!
>
> **Societal impact statement:** We strongly agree that the space of BO for drug design is one where societal impact is an important consideration. See A.6 for some initial discussion of societal impact. We are also happy to include an additional societal impact statement in the main text for the camera ready version and mention the dual use nature of such methods, e.g. Urbina et al., 2022.

---

### Meta-Review · Area_Chair_FmiF · 2022-08-27

**Recommendation:** Accept
**Confidence:** Certain

**Metareview:**

This paper develops a well-engineered approach to black-box optimization of expensive functions over structured spaces (e.g., graphs). The solution falls within the framework of Bayesian optimization (BO) over latent (continuous) space learned from unsupervised structures using deep generative models. One challenge in this framework is the high-dimensionality of the learned latent space. One straight-forward approach to handle this challenge is to apply state-of-the-art high-dimensional BO methods such as trust region based BO. The paper hypothesizes that there is a mismatch between the trust region in the latent space and the corresponding trust region in the structured space, and proposes a joint training/inference method over the Gaussian process surrogate model and the deep generative model to overcome this challenge. Experimental results on multiple molecule design optimization and arithmetic expression tasks show very good results over prior methods and shows improvements due to SELFIES representation for this approach.

Some of the reviewers' questioned the overall novelty of the approach as it combines advances in latent space BO, trust-region BO, and high-dimensional BO, but there is an agreement that overall strengths outweigh this concern. This paper makes a useful contribution to the latent space BO framework. Therefore, I recommend acceptance.

**Award:**

No

---

### Decision · Program_Chairs · 2022-09-14

Accept